# miR-100-5p Regulates Skeletal Muscle Myogenesis through the *Trib2*/mTOR/S6K Signaling Pathway

**DOI:** 10.3390/ijms24108906

**Published:** 2023-05-17

**Authors:** Kaiming Wang, Sui Liufu, Zonggang Yu, Xueli Xu, Nini Ai, Xintong Li, Xiaolin Liu, Bohe Chen, Yuebo Zhang, Haiming Ma, Yulong Yin

**Affiliations:** 1College of Animal Science and Technology, Hunan Agricultural University, Changsha 410128, China; m15116529648@163.com (K.W.); liufusui0816@163.com (S.L.); study236@163.com (Z.Y.); xxl1632501152@163.com (X.X.); 15291270712@163.com (N.A.); p13548614212@163.com (X.L.); lxl810711@163.com (X.L.); chenhe0213914@163.com (B.C.); ybzhangfd@126.com (Y.Z.); 2Guangdong Laboratory for Lingnan Modern Agriculture, Guangzhou 510642, China; 3Institute of Subtropical Agriculture, Chinese Academy of Sciences, Changsha 410125, China

**Keywords:** miR-100-5p, proliferation, differentiation, *Trib2*, mTOR, C2C12 myoblast

## Abstract

MicroRNAs (miRNAs) are endogenous small non-coding RNAs that play crucial regulatory roles in many biological processes, including the growth and development of skeletal muscle. miRNA-100-5p is often associated with tumor cell proliferation and migration. This study aimed to uncover the regulatory mechanism of miRNA-100-5p in myogenesis. In our study, we found that the miRNA-100-5p expression level was significantly higher in muscle tissue than in other tissues in pigs. Functionally, this study shows that miR-100-5p overexpression significantly promotes the proliferation and inhibits the differentiation of C2C12 myoblasts, whereas miR-100-5p inhibition results in the opposite effects. Bioinformatic analysis predicted that *Trib2* has potential binding sites for miR-100-5p at the 3′UTR region. A dual-luciferase assay, qRT-qPCR, and Western blot confirmed that *Trib2* is a target gene of miR-100-5p. We further explored the function of *Trib2* in myogenesis and found that *Trib2* knockdown markedly facilitated proliferation but suppressed the differentiation of C2C12 myoblasts, which is contrary to the effects of miR-100-5p. In addition, co-transfection experiments demonstrated that *Trib2* knockdown could attenuate the effects of miR-100-5p inhibition on C2C12 myoblasts differentiation. In terms of the molecular mechanism, miR-100-5p suppressed C2C12 myoblasts differentiation by inactivating the mTOR/S6K signaling pathway. Taken together, our study results indicate that miR-100-5p regulates skeletal muscle myogenesis through the *Trib2*/mTOR/S6K signaling pathway.

## 1. Introduction

Skeletal muscle is one of the most dynamic and plastic tissues in mammals; it accounts for about 40–50% of body mass in the adult stage; and plays a vital role in movement, respiration, and metabolism [1]. Skeletal muscle myogenesis is a strict and complex process through which mononuclear myoblasts undergo proliferation and differentiation, further fuse into multinuclear myotubes, and ultimately form muscle fibers with contractile traits [2]. This process is regulated by the cascade control of multiple transcription factors [3,4,5,6]. For example, the differentiation of myoblasts is activated by myogenic differentiation (*MyoD*), myogenin (*MyoG*), and myocyte enhancer factor 2 (*MEF2*). Non-coding RNAs also play crucial roles in the myogenesis process [7].

To date, emerging studies have suggested that non-coding RNAs affect skeletal muscle development by regulating gene expression, including microRNAs (miRNAs) [8], circular RNAs [9], and long non-coding RNAs (IncRNAs) [10]. Notably, miRNAs are a class of conserved small single-stranded RNAs that do not encode protein [11]. These miRNAs generally act as negative regulators of protein translation by influencing messenger RNA (mRNA) stability [12]. An increasing number of studies have suggested that miRNAs play a critical regulatory role in the process of skeletal muscle myogenesis. For example, miR-325-3p [13], miR-2400 [14], and microRNA-24-3p [15] were found to regulate skeletal growth and development. miR-100-5p is an important member of the miR-99 family (miR-99a, miR-99b, and miR-100) with the same seed region sequence [16]. Previous studies have reported that the miR-99 family plays a regulatory role in many kinds of cancer cells by targeting different genes, such as *IGF1R*, *AKT1*, and *mTOR* [17,18,19]. Notably, members of the miR-99 family are also involved in the regulation of myogenesis. For example, in a previous study, miR-99a-5p promoted the proliferation and inhibited the differentiation of skeletal muscle satellite cells by targeting *MTMR3* in chickens [20]. miR-100 overexpression could inhibit differentiation and promote intramuscular lipid deposition by modulating *IGF1R* in fetal bovine muscle satellite cells [17]. However, the effect of miR-100-5p in skeletal muscle myoblasts and its molecular regulatory mechanism remains unclear.

Previous studies have reported that miR-100 can bind directly to tribbles homolog 2 (*TRIB2*) 3′UTR in pulmonary fibroblasts [21]. *TRIB2* is a member of the tribbles family that functions as scaffold molecules (MAPK and Akt signaling pathways) in different cellular processes [22,23,24]. Additionally, as a cancer-associated pseudokinase, *TRIB2* can interact with E3 ubiquitin ligases and regulate the stability of downstream effectors, which impact various cellular processes such as proliferation, differentiation, migration, and cell death [25,26,27]. Recent studies have reported that *TRIB2* plays a critical role in vascular smooth muscle cell proliferation by modulating ERK activity [28] and that *TRIB2* is also involved in visceral fat accumulation [29]. However, to date, it remains unknown whether *TRIB2* plays a role in the growth and development of skeletal muscle.

The mammalian target of rapamycin (mTOR) is an evolutionarily conserved serine/threonine kinase complex that plays a critical role in regulating the development of various cells [30]. The mTOR complex has two protein complexes, mTOR complex 1 (mTORC1) and mTOR complex 2 (mTORC2). mTORC1 plays a dominant role in skeletal muscle development by activating the downstream ribosomal protein S6kin-1 (S6k1) and elF4-E binding protein 1 (4E-BP1), which affect protein synthesis [31]. During myoblast differentiation, the mTOR signaling pathway is activated by upstream IGF1R/PI3K/Akt signaling, resulting in increased phosphorylation of S6K and promoting mRNA translation and protein synthesis [32]. Therefore, activation of the mTOR signaling pathway is essential for skeletal muscle differentiation and regeneration [33].

In this study, we aimed to systematically investigate the molecular mechanism of miR-100-5p regulating skeletal muscle myogenesis. A series of experiments demonstrated that miR-100-5p promotes C2C12 myoblast proliferation by targeting *Trib2* and inhibits C2C12 myoblast differentiation through the *Trib2*/mTOR/S6K signaling pathway. This finding will contribute to expanding our understanding of muscle growth regulation.

## 2. Results

### 2.1. The Expression of miR-100-5p Is Associated with Myogenesis

Sequence alignment analysis revealed that miR-100-5p is highly conserved, with almost no differences between different species (Figure 1A). For the expression patterns of miR-100-5p in different pig tissues, qRT–PCR results showed that miR-100-5p is highly expressed in muscle tissue and spleen in piglets (Figure 1B). In addition, the expression trend of miR-100-5p in the four developmental stages of the *Longissimus dorsi* muscle was found to be dynamic (Figure 1C). These results suggest that miR-100-5p may also be expressed in myoblasts and exert certain influences. To explore the influence of miR-100-5p on skeletal muscle myogenesis, we selected mouse skeletal muscle C2C12 myoblasts because the mature sequences of miR-100-5p are highly conserved between pigs and mice. First, we detected the expression patterns of miR-100-5p in myoblasts using qTR-PCR. The results showed that the expression levels of the proliferation marker gene *Pcna* and miR-100-5p increased gradually with cell proliferation (Figure 1D,E). The expression levels of the differentiation marker gene *Myhc* gradually increased, while miR-100-5p gradually decreased with cell differentiation (Figure 1F,G). To investigate the function of miR-100-5p in myoblast proliferation and differentiation, miR-100-5p inhibitors, and mimics were transfected into myoblasts to inhibit and promote their expression levels, respectively (Figure 1H,I).

### 2.2. miR-100-5p Promotes Myoblast Proliferation and Inhibits Myoblast Differentiation

To investigate the effects of miR-100-5p on myoblast proliferation, we conducted miR-100-5p overexpression and inhibition experiments to detect the proliferation abilities of myoblasts. The EdU incorporation assay showed that the anti-miR-100-5p group significantly inhibited myoblast proliferation (Figure 2A,C), whereas miR-100-5p overexpression significantly promoted myoblast proliferation (Figure 2A,B). The CCK-8 assay found that cell proliferation in the miR-100-5p silence group was markedly inhibited (Figure 2D). In contrast, miR-100-5p overexpression promoted myoblast proliferation (Figure 2E). Next, the result of flow cytometry suggested that miR-100-5p knockdown increased the population of cells in the G1 phase and decreased the population of cells in the G2 phase and S phase (Figure 2F). After transfection with the miR-100-5p mimic, the number of cells in the G1 phase decreased while those in the G2 and S phases increased significantly (Figure 2G). Results of the qRT–PCR and Western blot showed that miR-100-5p knockdown or overexpression could regulate the mRNA and protein expression of proliferation-related genes (*Pcna* and *Cdk4*) (Figure 2H–J). Taken together, these results indicate that miR-100-5p promotes myoblast proliferation.

To further confirm miR-100-5p’s role in myoblast differentiation, we measured the expression of myogenic differentiation genes and fusion marker genes using qRT–PCR and Western blot. The qRT–PCR suggested that the mRNA expression levels of *Myod* and *MyoG* significantly increased after transfection with miR-100-5p inhibitor at 2 d, 4 d, and 6 d of differentiation (Figure 3A,B). Conversely, miR-100-5p overexpression significantly decreased mRNA expression levels of *Myod* and *Myog* (Figure 3A,B). Western blot analysis showed that the protein expression of MyoG and MyHC significantly increased following miR-100-5p knockdown (Figure 3C,D), whereas the protein expression of MyoG and MyHC decreased markedly after miR-100-5p overexpression (Figure 3E,F). Furthermore, qRT–PCR results showed that miR-100-5p knockdown remarkably promoted the mRNA expression level of *Myomaker* and *Myomixer* (Figure 3G), which are marker genes of muscle fusion, whereas miR-100-5p overexpression significantly decreased their mRNA expression levels (Figure 3H). These results suggest that miR-100-5p inhibits myoblast differentiation.

### 2.3. Target Gene Screening Revealed That miR-100-5p Directly Targeted Trib2

To explore the molecular mechanism of miR-100-5p’s impact on skeletal muscle myoblast development, potential target genes were predicted using the miRDB, Targetsan, StarBase, and miRcode (Figure 4A). Venn analysis was applied to obtain eight candidate target genes of miR-100-5p (Figure 4B). To determine which of these target genes were negatively regulated by miR-100-5p, qRT–PCR was used to detect mRNA expression levels of eight genes at the myoblast proliferation and differentiation stages. Muscleblind-like splicing regulator 1 (*Mbnl1*) and ribonucleoprotein PTB-binding 2 (*Raver2*) were not detected by qRT–PCR because they featured low expression in the myoblasts, while the other six genes could be detected. At the proliferation stage, qRT–PCR analysis showed that miR-100-5p overexpression significantly increased the mRNA expression level of *Cdk4* (proliferation marker gene) and decreased the expression levels of tribbles pseudokinase 2 (*Trib2*), myotubularin-related protein (*Mtmr3*), and adaptor-related protein-complex-1-associated regulatory protein (*Aplar*), while miR-100-5p knockdown caused the opposite results (Figure 4C). At the differentiation stage, miR-100-5p overexpression significantly decreased the mRNA expression level of *Myhc* (differentiation marker gene) and *Trib2*, whereas miR-100-5p inhibition caused the opposite results (Figure 4D). Taking these qRT–PCR results together, *Mtmr3* and *Trib2* were selected as candidate target genes for further study because their expression was negatively regulated by miR-100-5p in both the proliferation and differentiation stages. Notably, miR-100-5p had a stronger effect on *Trib2* than on *Mtmr3*.

To verify whether miR-100-5p can target the *Mtmr3* and *Trib2* genes, the putative sequences of the miR-100-5p binding site (WT) in *Mtmr3*/*Trib2* 3′UTR and the corresponding mutated sequences (MUT) were cloned into a pmirGLO Dual-Luciferase miRNA Target Expression Vector (Figure 4E,F). Then, the pmirGLO–*Mtmr3*/*Trib2*–WT and pmirGLO–*Mtmr3*/*Trib2*–MUT report vectors were co–transfected with miR-100-5p mimic or mimic NC into HEK293T cells. Our results confirmed that the miR-100-5p mimic failed to decrease the luciferase activity of pmirGLO–*Mtmr3*–MUT (Figure 4G). Conversely, miR-100-5p overexpression decreased the luciferase activity of pmirGLO–*Trib2*–MUT (Figure 4H). Therefore, *Trib2* was further identified as a possible target gene of miRNA-100-5p. Next, Western blot analysis showed that miR-100-5p could negatively regulate the protein expression of TRIB2 after miR-100-5p overexpression and knockdown at the myoblast differentiation stage (Figure 4I–K). These results indicated that *Trib2* is the direct target gene of miR-100-5p.

### 2.4. Trib2 Knockdown Promotes Myoblast Proliferation and Inhibits Myoblast Differentiation

We further examined the function of *Trib2* using specific RNA knockdown. We used the RNA-seq results from a previous study [34] and found that *TRIB2* expression was gradually up-regulated with chicken myoblast differentiation (Figure 5A). In addition, the result of qRT–PCR showed that the mRNA expression levels of *Myhc* and *Trib2* increased gradually with mouse C2C12 myoblast differentiation (Figure 5B,C). Immunofluorescence staining showed that *TRIB2* was localized to the nucleus and cytoplasm in differentiated C2C12 myoblasts (Figure 5D). To investigate the effects of *Trib2* on myoblast proliferation, Western blot analysis, and CCK-8 were performed. Western blot analysis showed that *Trib2* knockdown significantly decreased the protein expression of *TRIB2* while increasing the protein expression of PCNA (Figure 5E,F). The CCK-8 assay revealed that *Trib2* silencing significantly promoted myoblast proliferation 24 h, 48 h, and 72 h after transfection (Figure 5G). We also detected the influence of *Trib2* knockdown on myoblast differentiation. *Trib2* knockdown decreased the mRNA expression levels of *Trib2* and differentiation marker genes (*MyoG* and *MyHC*) (Figure 5H–J) and decreased the protein expression of MyoG and MyHC (Figure 5K,L). These results indicate that *Trib2* knockdown promotes myoblast proliferation and suppresses myoblast differentiation.

### 2.5. miR-100-5p Attenuates Activation of the mTOR Signaling Pathway during Myoblast Differentiation

Previous studies have reported that the activation of PI3K/Akt/mTOR signaling is indispensable for myogenic differentiation. Thus, we hypothesized that miR-100-5p might negatively regulate myoblast differentiation through the mTOR signaling pathway. To verify this hypothesis, we examined the protein expression of phosphor-mTOR (p-mTOR) and phosphor-S6K (p-S6K) after transfection with miR-100-5p inhibition, *Trib2* knockdown, and miR-100-5p overexpression in myoblasts using Western blot. The results showed that miR-100-5p knockdown significantly increased the protein expression of p-mTOR and p-S6K (Figure 6A,B), while *Trib2* knockdown and miR-100-5p overexpression markedly decreased the protein expression of p-mTOR and p-S6K (Figure 6C,D). Additionally, *Trib2* knockdown attenuated the effects of miR-100-5p inhibition on the protein expression of TRIB2, p-mTOR, p-S6K, and differentiation-related genes MyoG and MyHC (Figure 6E,F). To clarify the direct relationship between the mTOR signaling pathway and myoblast differentiation, an mTOR-specific inhibitors (Rapamycin) and its control group, dimethyl sulfoxide (DMSO), were used to treat myoblasts that were differentiated for four days. The results showed that Rapamycin significantly inhibited the protein expression of p-mTOR, p-S6K, TRIB2, and differentiation-related genes MyoG and MyHC (Figure 6G,H). Collectively, these data demonstrate that miR-100-5p is a negative regulator that inhibits myoblast differentiation and inactivates the mTOR signaling pathway through the downregulation of *Trib2* expression.

## 3. Discussion

In the orderly arranged myogenesis process, myoblasts undergo proliferation. Then, they exit the cell cycle and begin to differentiate. Finally, they fuse to form multinuclear fibers [35]. This process is regulated by many transcription factors, signaling pathways, and non-coding RNAs [5,36,37]. Currently, most studies on the miR-99 family focus on the development of cancer cells [16]. Recently, researchers have found that miR-99a-5p promotes proliferation and inhibits the differentiation of chicken skeletal muscle satellite cells [20]. Notably, miR-99a-5p and miR-100-5p belong to the miR-99 family, whose members have the same maturation sequences. Therefore, we speculated that miR-100-5p may have an effect on skeletal muscle myogenesis. In our study, we aimed to investigate the function and mechanism of miR-100-5p in myogenesis. First, the expression profile of miR-100-5p in pig tissues was detected. The results showed that miR-100-5p expression was the highest in muscle tissues and was dynamically expressed in the *Longissimus dorsi* muscle development stages. Hence, miR-100-5p was found to be a broad-spectrum miRNA, not a muscle-specific miRNA. However, this miRNA was highly expressed in muscle tissue, consistent with previous studies [38]. Then, the correlation between miR-100-5p and C2C12 myoblast proliferation and differentiation was further verified. The results indicated that the miR-100-5p expression level increased gradually with myoblast proliferation and decreased gradually with myoblast differentiation. This result suggests that the role of miR-100-5p may be different in the proliferation and differentiation stages. Additionally, miR-100-5p has a highly conservative sequence among different species (mouse, human, chicken, pig, and cattle). These results suggest that miR-100-5p might be a muscle-associated miRNA, similar to previously reported muscle-related miRNAs [39,40,41].

miR-100-5p was previously identified as a cancer-related miRNA that plays a crucial role in different cancer cells [42,43,44]. It was also reported that miR-100-5p is significantly correlated with fat formation and is a candidate regulator of intramuscular fat deposition [45]. Functional studies showed that miR-100 could inhibit 3T3-L1 cell differentiation by decreasing the expression of *mTOR* and *IGF1R* [46]. In this study, EdU staining and CCK-8 assays showed that miR-100-5p inhibition and overexpression, respectively, suppressed and promoted C2C12 myoblast proliferation. Flow cytometry analysis showed that miR-100-5p knockdown caused proliferating myoblasts to remain at the G1 phase, whereas the number of myoblasts in the S phase was significantly increased under miR-100-5p overexpression compared to the negative control group. The mRNA and protein expression of proliferation marker genes were also significantly changed when miR-100-5p was inhibited or overexpressed. Further, we found that miR-100-5p repressed C2C12 myoblast differentiation by detecting key regulators of muscle differentiation and fusion. This result is consistent with previous results where miR-100 inhibited the differentiation of bovine satellite cells [17]. In the myogenesis process, myoblasts exit the cell cycle and begin to differentiate [5]. Hence, the relationship between proliferation and differentiation is usually the opposite. Taken together, these findings suggest that miR-100-5p promotes the proliferation and inhibits the differentiation of C2C12 myoblast.

Most studies on miRNAs have concentrated on their capacity to regulate cellular processes by targeting different genes [11]. Therefore, the identification of miR-100-5p targets is vital for a comprehensive understanding of miRNA-mediated gene regulation [12]. Previous reports have shown that miR-100-5p regulates cell biological processes by targeting different genes. In this study, eight candidate target genes of miR-100-5p were identified via bioinformatics analysis (miRDB, starBase, miRcode, and Targetscan). Among these genes, the mRNA expression of *Trib2* and *Mtmr3* was negatively regulated by miR-100-5p in both the C2C12 myoblast proliferation and differentiation stages. A dual-luciferase reporter assay is the most direct and important means to validate the targeting relationship between miRNA and its target genes [47]. The results showed that miR-100-5p could directly target *Trib2*, which is consistent with previous reports that miR-100-5p and *TRIB2* have a targeted regulatory relationship [21]. Regrettably, miR-100-5p cannot bind to *Mtmr3*, which is a muscle-associated gene [20]. In addition, the protein expression of *TRIB2* was found to be regulated after miR-100-5p overexpression or inhibition. Ultimately, we confirmed that *Trib2* is the most likely target gene of miR-100-5p during C2C12 myoblast proliferation and differentiation.

*Trib2* is a pseudokinase associated with cancer and can interact with E3 ubiquitin ligase to control downstream effector protein stability [48,49,50]. *Trib2* is also involved in the MAPK and AKT pathways [51]. To date, the regulation of *Trib2* in skeletal muscle development has not been reported. Given the function of *Trib2*, we speculated that *Trib2* may play a role in myoblast proliferation and differentiation. First, the expression of *Trib2* gradually increased with myoblast differentiation, consistent with the previous reports on chicken myoblasts [34] and contrary to results for miR-100-5p. *Trib2* was localized in the nucleus and cytoplasm of differentiated myoblasts. Functionally, siRNA was used to knockdown *Trib2* expression in the myoblasts. Then, myoblast proliferation was significantly promoted, and myoblast differentiation was inhibited, which is consistent with miR-100-5p overexpression. In conclusion, miR-100-5p regulates myoblast proliferation and differentiation by targeting *Trib2*.

Activation of the mTOR signaling pathway is essential for muscle differentiation and regeneration [52]. mTOR is activated by the upstream IGF1R/PI3K/Akt pathway, resulting in increased phosphorylation of S6K for promoting mRNA translation and protein synthesis [53]. miR-100-5p was found to target several key genes in this signaling pathway, such as *IGF1R*, *AKT1*, and *mTOR* [18]. This study demonstrated the targeting relationship between miR-100-5p and *Trib2*. It was already acknowledged that *Trib2* is a protein scaffold for the AKT and MAPK pathways [23]. Hence, we speculated that miR-100-5p may ultimately regulate skeletal muscle development through the mTOR signaling pathway. C2C12 myoblasts were transfected with miR-100-5p overexpression, miR-100-5p inhibition, and *Trib2* knockdown to detect the phosphorylated protein expression of mTOR and S6K. The results showed that miR-100-5p and *Trib2* knockdown could inactivate the mTOR signaling pathway. Further experiments showed that miR-100-5p could target *Trib2* to inactivate the mTOR signaling pathway, thereby inhibiting myoblast differentiation. These results are consistent with previous reports that mTOR signaling pathway activation is vital for muscle differentiation [54,55,56]. Taken together, these results clearly indicate that miR-100-5p affects the mTOR signaling pathway by regulating the *Trib2* gene in myoblast differentiation.

## 4. Materials and Methods

### 4.1. Cell Culture

HEK293T cells and C2C12 myoblasts (ATCC, New York, NY, USA) were cultured in a growth medium with high-glucose DMEM (Giboc) containing 10% fetal bovine serum (Giboc) in a cell incubator with a humid environment of 37 °C and 5% CO_2_. The C2C12 cells of between 10 and 25 generations were stimulated with DMEM containing 2% horse serum (Giboc) for differentiation when the degree of cell fusion reached 80%.

### 4.2. RNA Oligonucleotides and Cell Transfection

To explore the effects of miR-100-5p and its target gene on skeletal muscle myogenesis, the miR-100-5p inhibitor, an inhibitor negative control (inhibitor NC), miR-100-5p mimic, negative control (mimic NC or siRNA NC), and *Trib2* small interfering RNAs (siRNAs) were synthesized from GenePharma Co., Ltd. (Suzhou, China). All transient transfections in C2C12 myoblasts were performed with a lipofectamine 2000 regent (Invitrogen, Waltham, MA, USA) according to the manufacturer’s instructions. For proliferation experiments, C2C12 myoblast transfection was performed when cell density reached 40%. After 6 h of transfection, the medium was replaced with a growth medium. In the differentiation experiments, transfection was performed when the cell density reached 80%. After 6 h of transfection, the medium was replaced with a differentiation medium. All RNA oligonucleotides are listed in Table 1.

### 4.3. RNA Isolation and Real-Time Quantitative PCR

Total RNA was extracted from tissues and myoblasts using TRIzol reagent (Invitrogen, USA) and stored at −80 °C. Total RNAs were reverse-transcribed into cDNA using a Thermo Scientific™ RevertAid™ First Strand cDNA Synthesis Kit. The specific stem-loop of miR-100-5p was used to synthesize the first strand cDNA for miR-100-5p. qRT–PCR was performed using an SYBR Green Kit (TransGen, Beijing, China). GAPDH was used to normalize mRNA expression, and U6 was used to normalize miR-100-5p expression. The relative expression of miR-100-5p and related genes was calculated via the 2^−ΔΔCt^ method. All primers are listed in Table 2.

### 4.4. Cell Counting Kit-8 (CCK-8) Assays

C2C12 myoblasts were seeded at a density of 1 × 10^5^ in 96-well plates. Cells were transfected with RNA Oligonucleotides when cell confluence reached 30%. After 12 h, 24 h, 48 h, and 72 h of culturing at 37 °C. Then, the cells were switched to the medium with 10% CCK-8 regent (AFExBIO, Houston, TX, USA) and incubated for 4 h at 37 °C. The cell absorbance value at 450 nm was detected using a spectrometer (Molecular Devices, San Francisco, CA, USA).

### 4.5. 5-Ethynyl-20-Deoxyuridine (EdU) Assay

To assess the effects of miR-100-5p and its target on myoblast proliferation, when C2C12 myoblast confluence reached 70–80%, cells were switched to a culture medium with EdU (20 µM) for 2 h at 37 °C. Then, EdU staining was performed using an EdU cell Proliferation Kit with Alexa Fluor 555 (Meilun, Dalian, China) following the manufacturer’s protocol. Images were captured using a fluorescence microscope (Zeiss LSM800, Carl Zeiss AG, Oberkochen, Germany). Finally, the ratio of EdU-positive cells was calculated as follows: (EdU-positive cells/DAPI staining cells) × 100%.

### 4.6. Immunofluorescence Staining

The differentiated myoblasts were fixed in 4% paraformaldehyde for 30 min and then permeabilized in 0.5% Triton X-100 for 20 min. Subsequently, the myoblasts were blocked with 5% BSA for 30 min. Immunofluorescence staining was performed using a primary antibody working solution overnight at 4 °C. Then, the second antibody working solution was added and incubated at room temperature for 1 h, and DAPI (1:1000) was incubated in the dark for 10 min. Images were obtained using a Leica SP8 confocal microscope (eyepiece magnification: 10×) and processed using Image J.

### 4.7. Flow Cytometric Analysis

The myoblasts were seeded in a 6-well plate. After 48 h of transfection, the myoblasts were washed three times with PBS. Then, the cells were collected and fixed in 75% ethanol at −20 °C overnight. The myoblasts were stirred with a PI staining solution (Yeasen Biotechnology, Shanghai, China) at 37 °C for 30 min. The myoblasts were then suspended and detected via flow cytometry (Becton Dickinson, Trenton, NJ, USA) according to the manufacturer’s protocols.

### 4.8. Dual-Luciferase Reporter Assay

A Dual-Luciferase Reporter Assay was employed to verify the targeting relationship between miR-100-5p and its candidate targeting genes (*Mtmr3* and *Trib2*). The putative sequences of the miR-100-5p binding site in *Mtmr3*/*Trib2* 3’UTR and the corresponding mutated sequences were cloned into a pmirGLO Dual-Luciferase miRNA Target Expression Vector (Promega, Madison, WI, USA). The pmirGLO–*Mtmr3*/*Trib2*–WT and pmirGLO–*Mtmr3*/*Trib2*–MUT report vector was co–transfected with the miR-100-5p mimic or mimic NC into HEK293T cells. After 24 h of transfection, the Dual-Luciferase Reporter Assay was performed using a Dual-Luciferase Reporter Assay System (Promega, Fitchburg, WI, USA) to detect the luciferase activity according to the manufacturer’s instructions.

### 4.9. Western Blot Assay

The total protein was extracted using a RIPA-lysed buffer with protease inhibitors and protein phosphatase inhibitors (Abiowell, Changsha, China). Then, the protein concentration was measured using a BCA Protein Assay Kit (Abiowell, Changsha, China). We performed Western blotting using various antibodies according to the standard procedure. The Datails of antibodies are as follows: PCNA 1:2000 (AF02986, Rabbit, AFBio, Changsha, China), CDK4 1:2000 (AF06640, Rabbit, AFBio, Changsha, China), MyHC 1:2000 (MF 20, Mouse, DSHB, Iowa City, IA, USA), MyOG 1:1000 (AB2146602, Mouse, DSHB, Iowa City, IA, USA), TRIB2 1:2000 (DF2692, Rabbit, Affinity Biosciences, Jiangsu, China), p-mTOR Ser2448 1:1000 (381557, Rabbit, ZenBio, Chengdu, China), p-S6K Thr389 1:1000 (9234T, Rabbit, Cell Signaling Technology, Danvers, MA, USA), and β-actin 1:5000 (AWS0001, Mouse, Proteintech, San Diego, CA, USA).

### 4.10. Bioinformation Analysis

Target gene predictions for miR-100-5p were performed using miRDB (http://www.miRbase.org/cgi-bin/broe.pl, accessed on 1 September 2022), Targetsan (http://www.targetscan.org/, accessed on 1 September 2022), StarBase (https://starbase.sysu.edu.cn/, accessed on 1 September 2022), and miRcode (http://www.mircode.org/, accessed on 1 September 2022). The Venn analysis was performed using an online tool (https://bioinfogp.cnb.csic.es/tools/venny, accessed on 1 September 2022).

### 4.11. Statistical Analysis

Image J was used to count EdU-positive cells and analyze protein bands. The relative mRNA expression under real-time quantitative PCR (qRT–PCR) was calculated using 2^−ΔΔCT^. An unpaired Student’s *t*-test was performed for the treatment and control groups using GraphPad Prism 8. These data are presented as the means ± standard error of the mean (SEM, n = 3). A difference was considered significant when the *p*-value was <0.05.

## 5. Conclusions

In summary, our study indicates that miR-100-5p promotes C2C12 myoblast proliferation by targeting *Trib2* and inhibits C2C12 myoblast differentiation by the *Trib2*/mTOR/S6K signaling pathway (Figure 7).

## Figures and Tables

**Figure 1 ijms-24-08906-f001:**
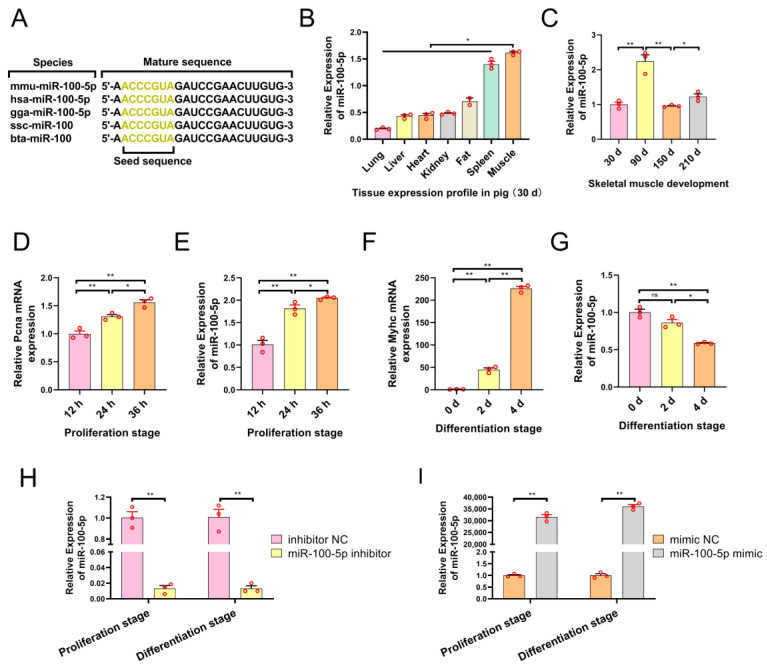
The expression of miR-100-5p is associated with myogenesis. (**A**) Conservation of the miR-100-5p sequence in different species. (**B**) Expression levels of miR-100-5p in different pig tissues. (**C**) Expression levels of miR-100-5p in *Longissimus dorsi* muscles at four developmental stages in pigs. (**D**,**E**) Expression levels of the proliferation marker gene *Pcna* and miR-100-5p in the C2C12 myoblast proliferation stage. (**F**,**G**) Expression levels of the differentiation marker gene *Myhc* and miR-100-5p in the C2C12 myoblast differentiation stage. (**H**) The transfection efficiency of miR-100-5p inhibition. (**I**) The transfection efficiency of miR-100-5p overexpression. The results are presented as the mean ± SEM (n = 3). * *p* < 0.05; ** *p* < 0.01; *p* ≥ 0.05: ns (Not significant).

**Figure 2 ijms-24-08906-f002:**
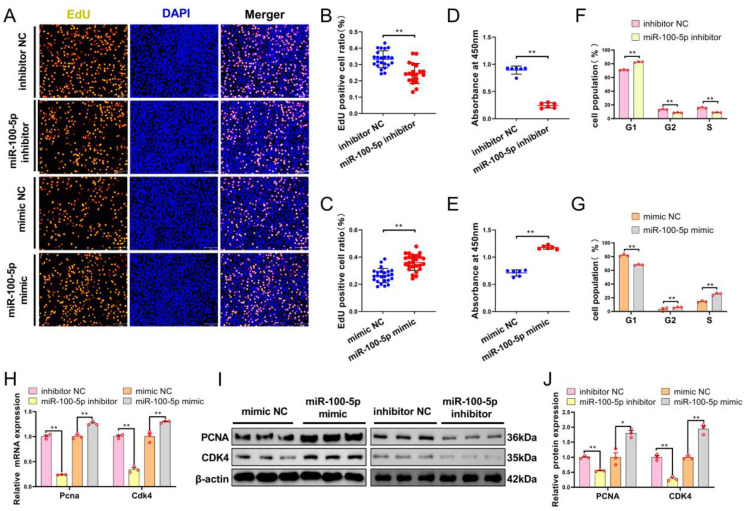
miR-100-5p promotes myoblast proliferation. (**A**–**C**) EdU staining detected the proliferation of myoblasts transfected with the miR-100-5p inhibitor and mimic. Cells in the DNA replication phase were stained with EdU (green), and nuclei were stained with DAPI (blue). Scale bar = 50 μm. The proportion of EdU-positive cells among the total cells was calculated. (**D**,**E**) CCK-8 assay for myoblasts after miR-100-5p overexpression and inhibition. (**F**,**G**) Cell-cycle analysis, using propidium iodide staining, of myoblasts transfected with the miR-100-5p inhibitor and mimic. (**H**–**J**) qRT–PCR and Western-blot-detected expression of the proliferation-related gene after transfection with the miR-100-5p inhibitor and mimic. The results are presented as the mean ± SEM (n = 3). * *p* < 0.05; ** *p* < 0.01.

**Figure 3 ijms-24-08906-f003:**
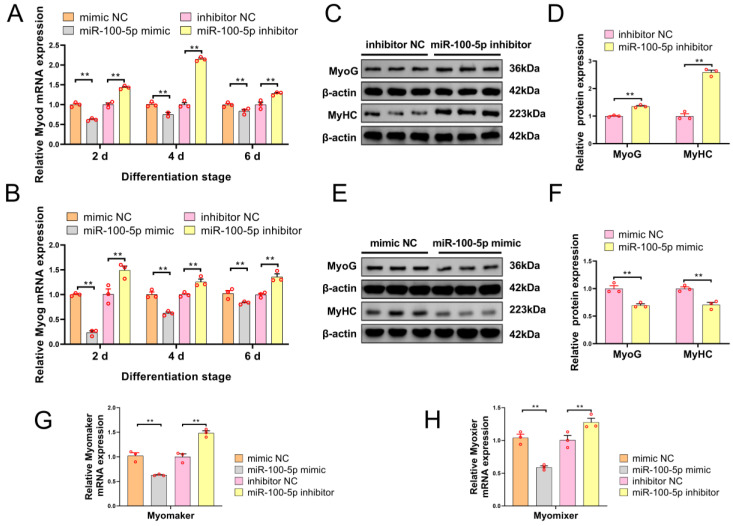
miR-100-5p inhibits myoblast differentiation. (**A**,**B**) qRT–PCR-detected mRNA expression levels of myogenic differentiation (*Myod*) and Myogenin (*Myog*) after overexpression and inhibition of miR-100-5p in the differentiation stage. (**C**–**F**) Western-blot-detected protein expression of Myogenin (MyoG) and Myosin heavy chain (MyHC) in myoblasts after transfection with the miR-100-5p inhibitor and miR-100-5p mimic, respectively. (**G**,**H**) The mRNA expression levels of muscle fusion marker genes (*Myomaker* and *Myomixer*) were detected by qRT–PCR in myoblasts after miR-100-5p overexpression and inhibition. The results are presented as the mean ± SEM (n = 3). ** *p* < 0.01.

**Figure 4 ijms-24-08906-f004:**
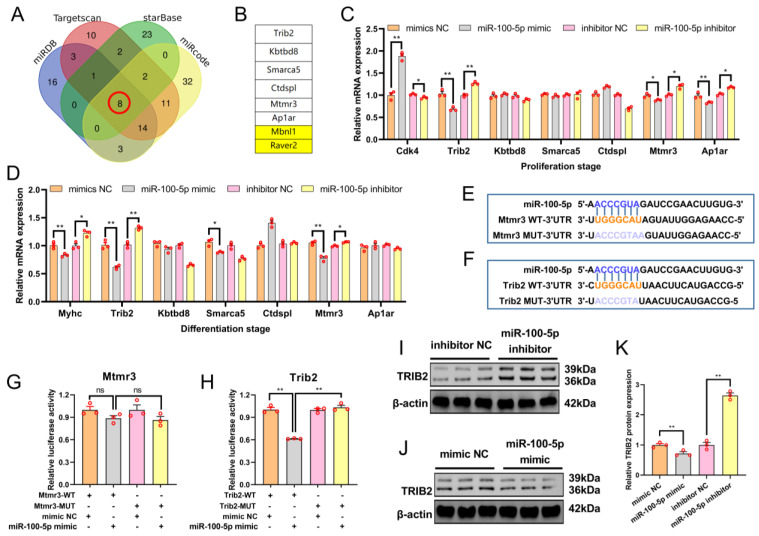
Target gene screening revealed that miR-100-5p directly targeted *Trib2*. (**A**) The target genes of miR-100-5p were predicted using miRDB, Targetsan, StarBase and miRcode. (**B**) Eight candidate target genes of miR-100-5p were obtained via Venn analysis. (**C**) The mRNA expression levels of proliferation marker gene *Cdk4* and six candidate target genes were detected by qRT–PCR after miR-100-5p overexpression and inhibition at the proliferation stage in myoblasts. (**D**) The mRNA expression levels of the differentiation marker gene *Myhc* and six candidate target genes were detected by qRT–PCR in myoblasts after miR-100-5p overexpression and inhibition at the differentiation stage. (**E**,**F**) Wild-type binding sites (WT) and mutation binding sites (MUT) of miR-100-5p in the *Mtmr3* 3′UTR and *Trib2* 3′UTR regions. (**G**,**H**) In the dual-luciferase reporter assay, the pmirGLO–*Mtmr3*/*Trib2*–WT and pmirGLO–*Mtmr3*/*Trib2*–MUT report vectors were co–transfected with the miR-100-5p mimic or mimic NC into HEK293T cells. (**I**–**K**) Western-blot-detected protein expression of TRIB2 in myoblasts after transfection with miR-100-5p inhibitor and miR-100-5p mimic. The results are presented as the mean ± SEM (n = 3). * *p* < 0.05; ** *p* < 0.01; *p* ≥ 0.05: ns (Not significant).

**Figure 5 ijms-24-08906-f005:**
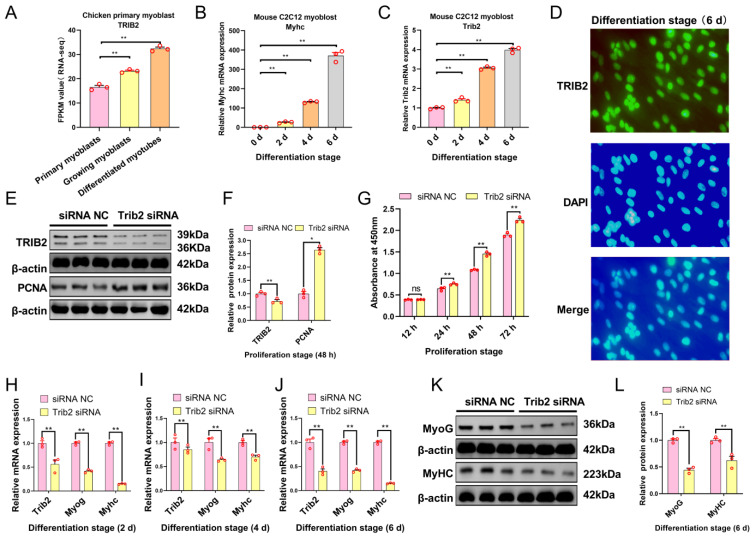
*Trib2* knockdown promotes myoblast proliferation and inhibits differentiation. (**A**) *TRIB2* expression was gradually up-regulated during chicken myoblast differentiation. (**B**,**C**) qRT–PCR detected the mRNA expression levels of *Myhc* and *Trib2* during C2C12 myoblast differentiation. (**D**) Differentiated mouse C2C12 myoblasts were stained with the TRIB2 antibody (green) and DAPI (blue); scale bar = 50 μm. (**E**,**F**) Western-blot-detected protein expression of TRIB2 and proliferation marker gene (PCNA) after transfection with *Trib2* siRNA in the myoblasts. (**G**) Cell counting detected by CCK-8 during myoblast proliferation. (**H**–**J**) The mRNA expression levels of *Trib2* and muscle differentiation maker genes (*Myog* and *Myhc*) were detected via qRT–PCR in myoblasts after being transfected with *Trib2* siRNA. (**K**,**L**) Western-blot-detected protein expression of differentiation marker genes (MyoG and MyHC) in myoblasts after transfection with *Trib2* siRNA. The results are presented as the mean ± SEM (n = 3). * *p* < 0.05; ** *p* < 0.01; *p* ≥ 0.05: ns (Not significant).

**Figure 6 ijms-24-08906-f006:**
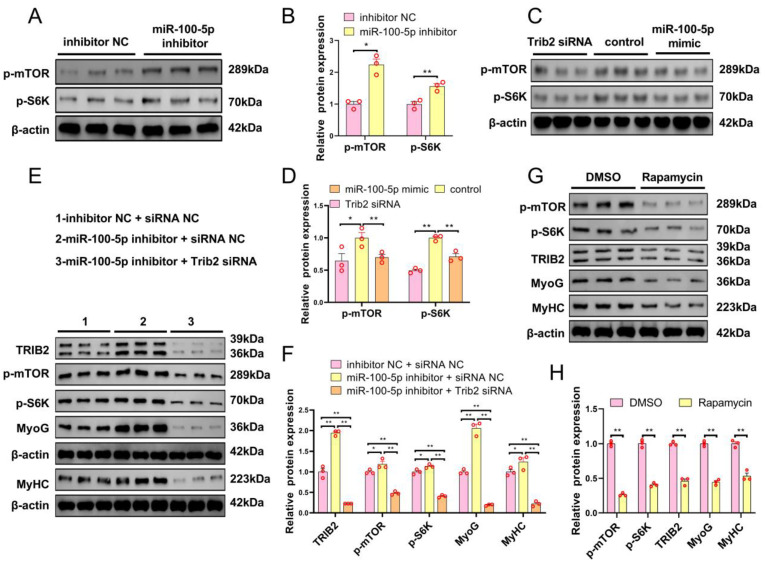
miR-100-5pmiR-100-5p inactivates the mTOR signaling pathway through downregulation of *Trib2* expression during myoblast differentiation. (**A**,**B**) Effect of miR-100-5p inhibition on the protein expression of phosphor-mTOR (p-mTOR) and phosphor-S6K (p-S6K) during myoblast differentiation. (**C**,**D**) Effects of miR-100-5p overexpression and *Trib2* knockdown on the protein expression of phosphor-mTOR (p-mTOR) and phosphor-S6K (p-S6K) during myoblast differentiation. (**E**,**F**) miR-100-5p inhibitor and *Trib2* siRNA were co-transfected into myoblasts. Western-blot-detected the protein expression of TRIB2, phosphor-mTOR (p-mTOR), phosphor-S6K (p-S6K), and differentiation-related genes (MyoG and MyHC). (**G**,**H**) Western-blot-detected the protein expression of phosphor-mTOR (p-mTOR), phosphor-S6K (p-S6K), TRIB2, and differentiation-related genes (MyoG and MyHC) in differentiated myoblasts that were treated with mTOR-pathway-specific inhibitors (Rapamycin) and the corresponding control group treated with dimethyl sulfoxide (DMSO). The results are presented as the mean ± SEM (n = 3). * *p* < 0.05; ** *p* < 0.01.

**Figure 7 ijms-24-08906-f007:**
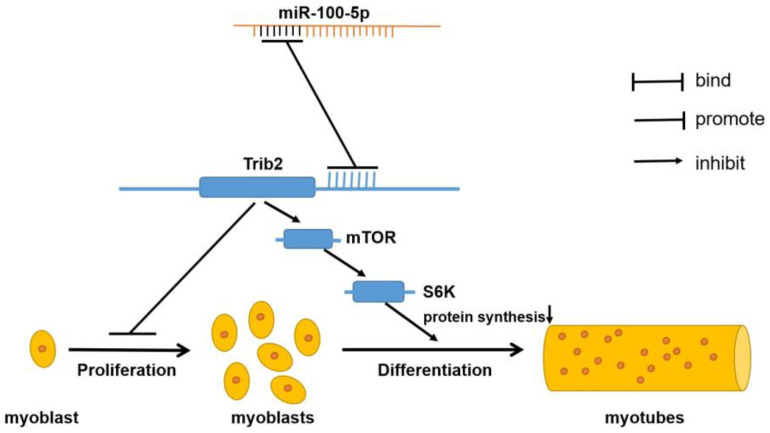
Mechanism diagram of miR-100-5p regulating skeletal muscle myogenesis.

**Table 1 ijms-24-08906-t001:** RNA oligonucleotides used in this study.

Name	Sequence (5′ to 3′)
miR-100-5p mimic	AACCCGUAGAUCCGAACUUGUG
	CAAGUUCGGAUCUACGGGUUUU
Mimic NC	UUCUCCGAACGUGUCACGUTT
	ACGUGACACGUUCGGAGAATT
miR-100-5p inhibitor	CACAAGUUCGGAUCUACGGGUU
Inhibitior NC	CAGUACUUUUGUGUAGUACAA
*Trib2* siRNA	GCCAGAGUUUCAGCCCGAATT
	UUCGGGCUGAAACUCUGGCTT
siRNA NC	UUCUCCGAACGUGUCACGUTT
	ACGUGACACGUUCGGAGAATT

**Table 2 ijms-24-08906-t002:** Primer information for miRNA and mRNA quantitative reverse transcription.

Gene	Primer Name	Primer Sequence (5′ to 3′)
*miR-100-5p*	Stem-loop	CTCAACTGGTGTCGTGGAGTCGGCAATTCAGTTGAGCACAAGTT
	R	TCGGCAGGAACCCGTAGATCCG
	F	CTCAACTGGTGTCGTGGA
*U6*	R	AACGCTTCACGAATTTGCGT
	F	CTCGCTTCGGCAGCACA
*Trib2*	R	CACTCTTGTCTCCCGATGCC
	F	ACACGGTCCTCTCCTACTTCT
*Pcna*	R	ATTCACCCGACGGCATCTTT
	F	GAACCTCACCAGCATGTCCA
*Cdk4*	R	TCAGGTCCCGGTGAACAATG
	F	GCCAAAGGAAGGAGGTAAGGG
*Ccnd*	R	ATAGGAACACTGCGGGAGGT
	F	GCCAAAGGAAGGAGGTAAGGG
*MyHC*	R	GAGCCTCGATTCGCTCCTTT
	F	CGGTCGAAGTTGCATCCCT
*MyoG*	R	CTGGGAAGGCAACAGACAT
	F	CAATGCACTGGAGTTCGGT
*Myomixer*	F	CAAGAAGTTCAGGCTTCAGGTG
	R	CACTTCTGGGGGCCCAATC
*myomaker*	F	AGGGGTCCAGGATAAAAGGC
	R	GCCAAGCATTGTGAAGGTCG
*Gapdh*	R	TCCACCACCCTGTTGCTGTAG
	F	AGGGCATCCTGGGCTACACT

## Data Availability

Data presented in this study are available on request from the corresponding author.

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
