# Peer review of "miR-100-5p Regulates Skeletal Muscle Myogenesis through the Trib2/mTOR/S6K Signaling Pathway"

_ijms, 2023, doi:10.3390/ijms24108906_

Round 1
Reviewer 1 Report
In their manuscript titled “miR-100-5p Regulates Skeletal Muscle Myogenesis through Trib2/mTOR/S6K Signaling Pathway,” Wang et al. report that miR-100-5p, a miRNA, plays a role in vertebrate skeletal muscle development by promoting myoblast proliferation and concurrently inhibiting differentiation through regulation of Trib2 and the mTOR signaling pathway. They show that miR-100-5p is conserved across vertebrates and promotes myoblast proliferation, including genes known to regulate proliferation. They also see downregulation of differentiation markers such as MyHC and MyoG. Trib2, consistent with a previous study, is directly bound by miR-100-5p via it’s 3’-UTR. Experimental manipulation of both Trib2 and mTOR signaling produce similar phenotypes to inhibition or overexpression of miR-100-5p, indicating these components work in the same regulatory pathway. This study identifies another mode of regulation affecting myoblast differentiation and proliferation, and identifies a role for Trib2 in this process. The experiments are well planned and have matched controls, and the logic of the manuscript and conclusions is easy to follow. The manuscriopt will be of interest to the muscle and Trib2 communities.
I have the following recommendations to improve the manuscript:
1. It would be nice to include the datapoints on the bar charts or to show box plots in all figure panels, rather than just showing the mean plus SEM, which can obscure variation in the raw data.
2. In Figure 2A, from the DAPI images provided, the cell look >90% confluent. The details are not provided in the methods in enough detail to know at which confluence this experiment was performed. C2C12 cells will start to differentiate near confluence, even without switching medium, which could influence the rate of cell division and EdU incorporation rates. Could the authors please clarify how this experiment was done, and what was done to ensure treatment of actively proliferating myoblasts throughout the experiment?
3. In Figure 5F, it appears that the efficiency of Trib2 knockdown is low (only 20-30%). It is surprising the effects are so strong in the knockdown experiments with Trib2, given such a small decrease in overall protein. Is Trib2 known to by haploinsufficient, or can the authors discuss how such a low level of knockdown can cause this phenotype? Please also see line 216, where the text says that Trib2 knockdown increased TRIB2 expression, which does not seem to be the case.
4. In the data for the inhibition of differentiation, the authors could also quantify the number of myotubes formed per unit area on the slides they have, as I would expect there to be a decrease in the number of nuclei per myotube or in the number of myotubes formed if differentiation is indeed promoted/inhibited depending on the particular manipulation (Figure 3 & 5). This might also offer physiological relevance to the changes in gene expression observed in proliferation and differentiation markers, and if these levels of changes are able to actually prevent or promote fusion and myotube formation.
5. The Materials and Methods are very cursory and offer few details on experimental procedures. It would not be possible to replicate some of the experiments based on what is presented.
a. For example, for section 4.1, what passage of C2C12 was used (C2C12 cells lose the capacity to differentiate in later passages, in our hands between 20-30)? What was the plating density used for experiments? At what confluence were experimental manipulations performed?
b. In section 4.3, which RT kit was used (Invitrogen has multiple kits)?
c. In 4.6, which confocal microscope and with which objectives?
d. In 4.8 the authors need to provide details on how the luciferase reporter constructs were cloned. It is not clear what WT and MUT refer to, as this region of the text (Line 178-182) is confusing. What exactly was mutated, and how was this generated? Which vector was used for generating the luciferase constructs? The authors should also provide the clone number or reagent ID for the antibodies they list in lines 416-420 in this section, as it is not even mentioned which species they are generated in to be able to order the same reagent, and in some cases, there are multiple antibodies from the cited company targeting that antigen.
6. Please read through the text with attention to grammar. The organization of the text and the logic flow are overall good and understandable. However, there are many small grammar mistakes. For example, there are places where a period is used when a comma is appropriate. The first letter of a sentence needs to be capitalized. There are also multiple “sentences” in the text that lack a verb, and thus are not actually sentences. There are also several spelling mistakes or possible “auto-correct” mistakes where the wrong word is used.
7. There is a study that just came out identifying a role for Trib2 in smooth muscle that should also be included in the revised version. DOI: 10.1002/cbin.11982
Please read through the text with attention to grammar. The organization of the text and the logic flow are overall good and understandable. However, there are many small grammar mistakes. For example, there are places where a period is used when a comma is appropriate. The first letter of a sentence needs to be capitalized, and notably in sections 2.3 and in the methods 4.8 there were more mistakes than in other sections. There are also multiple “sentences” in the text that lack a verb, and thus are not actually sentences. There are also several spelling mistakes or possible “auto-correct” mistakes where the wrong word is used.
Reviewer 2 Report
This manuscript deals with the role of MiR-100-5p and of its target Trib2 mRNA in promoting myoblast proliferation and inhibiting myoblast differentiation. Most work was carried out on C2C12 myoblasts, with a number of data were obtained in pig tissues. Further evaluation on the role of Trib2 in myogenesis showed that it affected the mTOR/S6K pathway.
In overall, the experiments were appropriate and the results correctly interpreted.
All the paper is focused on MiR-100-5p and Trib2, which is ok, however, the co-occurrence of other molecules in the complex signaling leading to the development of the different stages of myogenesis should be considered, at least in the Introduction and in the Discussion. I suggest adding appropriate references when quoting previous work on general issues.
English language is rather poor, with plenty of grammatical errors, misspelling, inaccuracies and sentences difficult to understand and it should be corrected with the aid of an expert.
In revising the manuscript, please notice that “MiR-100-59 mRNA expression” (Figure1) is a nonsense, since microRNAs are not mRNAs. Figures are somehow overcrowded and difficult to read. Legends on the Y axes have different size and legends on the X axes and on titles are sometimes halved, as if the text box was not high enough. In Fig. 5D an arrow points to nothing.
English language is rather poor, with plenty of grammatical errors, misspelling, inaccuracies and sentences difficult to understand and it should be corrected with the aid of an expert.
Round 2
Reviewer 1 Report
The authors have addressed the comments I raised on my previous review. The methods are much more thorough, which helps with interpretation of the results. The variability of the data is also much clearer with individual data points on the bar plots.
Grammar has been much improved. There are still a few spelling mistakes in edited areas, that will be easy to address in the final copy editing.